# RefinedFields: Radiance Fields Refinement for Planar Scene Representations

**Karim Kassab**  *k.kassab@criteo.com*
*Criteo AI Lab, Paris, France*
*LASTIG, Université Gustave Eiffel, IGN-ENSG, F-94160 Saint-Mandé*

**Antoine Schnepf**  *a.schnepf@criteo.com*
*Criteo AI Lab, Paris, France*
*Université Côte d'Azur, CNRS, I3S, France*

**Jean-Yves Franceschi**  *jycja.franceschi@criteo.com*
*Criteo AI Lab, Paris, France*

**Laurent Caraffa**  *laurent.caraffa@ign.fr*
*LASTIG, Université Gustave Eiffel, IGN-ENSG, F-94160 Saint-Mandé*

**Jeremie Mary**  *j.mary@criteo.com*
*Criteo AI Lab, Paris, France*

**Valerie Gouet-Brunet**  *valerie.gouet@ign.fr*
*LASTIG, Université Gustave Eiffel, IGN-ENSG, F-94160 Saint-Mandé*

**Reviewed on OpenReview:** *https://openreview.net/forum?id=S6JpSsYBDZ*

## Abstract

Planar scene representations have recently witnessed increased interests for modeling scenes from images, as their lightweight planar structure enables compatibility with image-based models. Notably, K-Planes have gained particular attention as they extend planar scene representations to support in-the-wild scenes, in addition to object-level scenes. However, their visual quality has recently lagged behind that of state-of-the-art techniques. To reduce this gap, we propose RefinedFields, a method that leverages pre-trained networks to refine K-Planes scene representations via optimization guidance using an alternating training procedure. We carry out extensive experiments and verify the merit of our method on synthetic data and real tourism photo collections. RefinedFields enhances rendered scenes with richer details and improves upon its base representation on the task of novel view synthesis. Our project page can be found at `https://refinedfields.github.io` .

## 1 Introduction

To draw novel objects and views, humans often rely on a blend of **cognition** and **intuition**, where the latter is built on a large prior acquired from a long-term continuous exploration of the visual world. Nevertheless, ablating one of these two elements results in catastrophic representations. On the one hand, humans find it particularly difficult to draw a bicycle based solely on this preconceived prior (Gimini, 2016). However, once one photograph is observed, drawing novel views of a bicycle becomes straightforward. On the other hand, drawing monuments and complex objects based solely on observed images, and with no preconceived notions of geometry and physics, is also non-trivial. In computer vision, recent methods tackling object generation and novel view synthesis generally focus on either the former or the latter ablation.

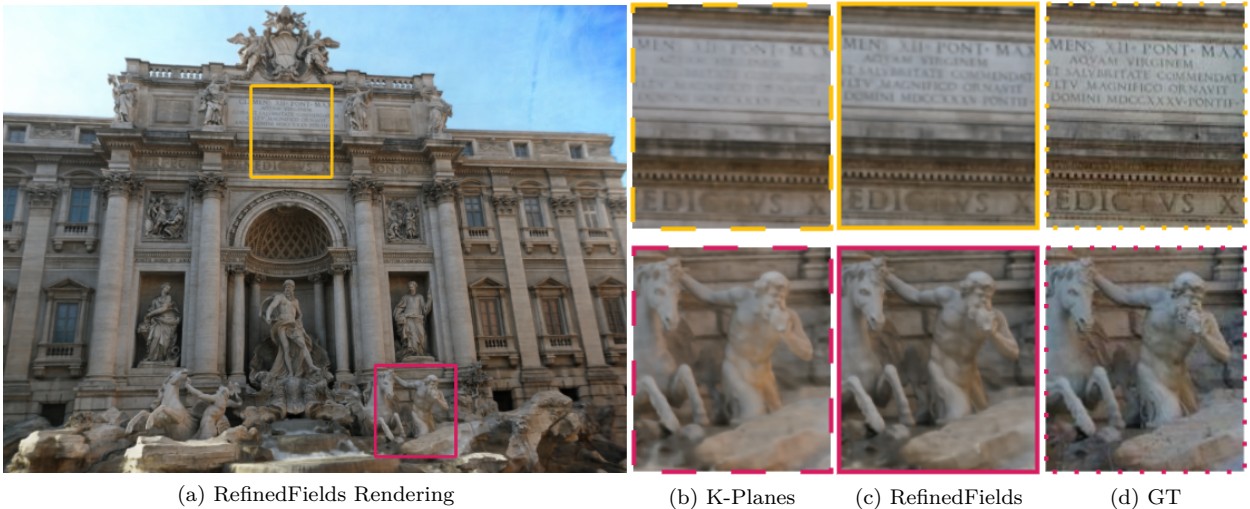

|  | (a) RefinedFields Rendering | | (b) K-Planes | (c) RefinedFields | (d) GT |

Figure 1: **Qualitative Results.** Given images of the Trevi fountain from Phototourism (Jin et al., 2020), as well as a pre-trained model (Rombach et al., 2022), our method leverages the pre-trained model and refines K-Planes with finer details that are under-represented when optimizing the same K-Planes on the images alone.

The first class of methods approaches novel view synthesis by learning scenes through rigorous **cognition**, as in dense observations of captured images. Although classic methodologies like structure-from-motion (Hartley & Zisserman, 2004) and image-based rendering (Shum et al., 2008) have previously tackled this problem, the field has recently seen substantial advancements thanks to stereo reconstruction techniques (Wang et al., 2024), as well as implicit neural representation methods (Mildenhall et al., 2020) which this paper focuses on. Particularly, planar scene representations such as Tri-Planes (Chan et al., 2022) have recently gained significant attention (Shue et al., 2023; Lan et al., 2024) for their lightweight planar structure that allows for a seamless integration with image-based models. Recent methods extend NeRFs (Martin-Brualla et al., 2021) and Tri-Planes (Fridovich-Keil et al., 2023, K-Planes) to support learning from unconstrained "in-the-wild" photo collections by enabling robustness against illumination variations and transient occluders. These representations, however, do not learn any prior across scenes as they are trained from scratch for each scene. This means that these representations are learned in a closed-world setting, where the information scope is limited to the training set at hand. Despite the wide interest in planar scene representations, their visual quality currently falls short behind recent state-of-the-art methods (Kulhanek et al., 2024; Xu et al., 2024). Our goal is to integrate pre-trained networks into the training framework of K-Planes, with an aim to improve their rendering performance.

The second class of methods tackles novel view synthesis and object generation by learning and leveraging priors over images and scenes, reminiscent of drawing from insights and **intuition**. These methods have also recently witnessed accelerated advancements. Recent works leverage pre-trained networks to achieve Object Generation (OG) (Poole et al., 2023; Metzer et al., 2023) and Novel View Synthesis (NVS) (Yu et al., 2021; Jain et al., 2021; Liu et al., 2023; Melas-Kyriazi et al., 2023). Particularly, Liu et al. (2023) achieve NVS from single images only by simply fine-tuning a pre-trained latent diffusion model (Rombach et al., 2022). This proves pivotal significance related to large-scale pre-trained vision models, as it shows that, although trained on 2D data, these models learn a rich geometric 3D prior about the visual world. Nevertheless, as these pre-trained models alone have no explicit multi-view geometric constraints, their use for 3D applications is usually prone to geometric issues (e.g. geometric inconsistencies, multi-face Janus problem, content drift issues (Shi et al., 2023, Figure 1)). This class of methods has not yet been explored to enhance implicit models representing in-the-wild scenes, as leveraging priors over such representations is not evident.

**RefinedFields proposal.** Our work builds on the previous discourse and aims to enhance planar scene modeling by leveraging pre-trained networks. To this end, we present an alternating training algorithm that

integrates a novel *scene refining* stage, which aims to propose a better optimization initialization by utilizing a pre-trained model. We adopt K-Planes (Fridovich-Keil et al., 2023) as a base scene representation, which extend Tri-Planes to support in-the-wild scenes, in addition to object-level scenes. Liu et al. (2024) highlight that features in planar scene representations resemble projected scene images, a finding we also corroborate in Appendix B, and exploit to improve upon K-Planes. Specifically, RefinedFields *refines* planar scene representations by projecting them onto the space of representations inferable by a pre-trained network, which pushes K-Planes features to more closely resemble real-world images. To do so, we build on the seamless integration of K-Planes with image-based models and present an alternating training procedure that iteratively switches between optimizing a K-Planes representation on images from a particular dataset, and fine-tuning a pre-trained network to output a new conditioning leading to a refined version of this K-Planes representation. Overall, this procedure guides the optimization of a particular scene, by leveraging not only the training dataset at hand but also the rich prior lying within the weights of the pre-trained model, which is a first for in-the-wild scene modeling.

We conduct extensive quantitative and qualitative evaluations of RefinedFields. We show that our method improves upon K-Planes with richer details in scene renderings. We prove via ablation studies that this added value indeed comes from the fine-tuned prior of the pre-trained network. Figure 1 illustrates the improvements our method showcases on the *Trevi fountain* scene from Phototourism (Jin et al., 2020).

## 2   Related Work

RefinedFields achieves geometrically consistent novel view synthesis, that can also be applied in-the-wild, by leveraging K-Planes, and a large-scale pre-trained network. Our method is the first to satisfy all of these attributes, as summarized in Table 1. In this section, we develop the various preceding works from which our method takes inspiration.

**Neural representations.**   Neural rendering (Tewari et al., 2020) has seen significant advancements since the introduction of Neural Radiance Fields (Mildenhall et al., 2020, NeRF). At its core, NeRF learns a scene by fitting the weights of a neural network on posed images of said scene. This subsequently enables the reconstruction of the scene thanks to volume rendering (Kajiya & Herzen, 1984). Subsequent to NeRFs, various scene representations have been introduced (Chen et al., 2022a; Müller et al., 2022; Kerbl et al., 2023). Particularly, Chan et al. (2022) introduce Tri-Planes, a planar scene representation serving as a middle ground between implicit and explicit representations, enabling a faster learning of scenes. Tri-Planes have been widely adopted in recent works (Shue et al., 2023; Lan et al., 2024; Wang et al., 2023b), as their planar structure enables a seamless integration with image-based models.

**Neural representations in-the-wild.**   Subsequent to NeRFs, several techniques emerged to extend the NeRF setup to "in-the-wild" unconstrained photo collections (Jin et al., 2020, Phototourism) plagued by illumination variations and transient occluders. This added variability makes learning a scene particularly challenging, as surfaces can exhibit significant visual disparities across views. NeRF-W (Martin-Brualla et al., 2021) addresses the challenge of novel view synthesis in-the-wild (NVS-W) by modeling scene lighting through appearance embeddings, and transient occluders through an additional transient head. Splatfacto-W (Xu et al., 2024) proposes a Nerfstudio (Tancik et al., 2023) implementation of 3D Gaussian Splatting (Kerbl et al., 2023, 3DGS) that extends standard 3DGS to support in-the-wild scene modeling via appearance embeddings. WildGaussians (Kulhanek et al., 2024) also proposes using 3DGS with appearance embeddings for in-the-wild scene modeling, but additionally predicts uncertainty masks to exclude transient occluders from the loss computation. It is important to note that although this work adopts a pre-trained network, it is solely used for the computation of the loss masks on the data, and not as a prior on the scene representation. This approach is orthogonal to our approach of leveraging a prior to directly improve upon a specific scene representations, as it acts on the data and not the scene representation itself. Fridovich-Keil et al. (2023) present K-Planes, which modify and extend Tri-Planes (Chan et al., 2022) to in-the-wild scenes thanks to learnable appearance embeddings, similarly to Martin-Brualla et al. (2021). Our work aims to improve upon K-Planes representations by extending their training beyond closed-world setups using pre-trained networks. Note that other works also tackle NVS-W: Chen et al. (2022b) model scene lighting through

Table 1: **Related work overview.** RefinedFields leverages a pre-trained prior (Rombach et al., 2022) on the scene representation to refine K-Planes, our underlying scene representation, utilized for novel view synthesis in the wild (NVS-W).
†Non exhaustive, other works with characteristics similar to NeRF-W exist. Refer to Section 2 for more details.

| | No 3D supervision | Pre-trained prior | Geometric consistency | In-the-wild scene modeling | Underlying representation | Task |
|---|---|---|---|---|---|---|
| NFD (Shue et al., 2023) | ✗ | ✗ | ✓ | ✗ | Tri-Planes | |
| 3DGen (Gupta et al., 2023) | ✗ | ✗ | ✓ | ✗ | Tri-Planes | OG |
| Latent-NeRF (Metzer et al., 2023) | ✓ | ✓ | ✓ | ✗ | NeRF | |
| DreamFusion (Poole et al., 2023) | ✓ | ✓ | ✓ | ✗ | NeRF | |
| NeRF (Mildenhall et al., 2020) | ✓ | ✗ | ✓ | ✗ | NeRF | |
| RealFusion (Melas-Kyriazi et al., 2023) | ✓ | ✓ | ✓ | ✗ | NeRF | |
| DiffusioNeRF (Wynn & Turmukhambetov, 2023) | ✓ | ✗ | ✓ | ✗ | NeRF | |
| NerfDiff (Gu et al., 2023) | ✗ | ✓ | ✓ | ✗ | Tri-Planes | NVS |
| 3DiM (Watson et al., 2023) | ✗ | ✗ | ✓ | ✗ | — | |
| Zero-1-to-3 (Liu et al., 2023) | ✗ | ✓ | ✗ | ✗ | — | |
| NeRF-W (Martin-Brualla et al., 2021) | ✓ | ✗ | ✓ | ✓ | NeRF | |
| WildGaussians (Kulhanek et al., 2024) | ✓ | ✗ | ✓ | ✓ | 3DGS | |
| Splatfacto-W (Xu et al., 2024) | ✓ | ✗ | ✓ | ✓ | 3DGS | NVS-W† |
| K-Planes (Fridovich-Keil et al., 2023) | ✓ | ✗ | ✓ | ✓ | K-Planes | |
| RefinedFields (ours) | ✓ | ✓ | ✓ | ✓ | K-Planes | |

appearance embeddings and transient occluders through transient embeddings, Yang et al. (2023) leverage interactive information across rays to mimic the perception of humans in-the-wild, and Chen et al. (2024) separate static and transient components by utilizing heuristics-guided segmentation. However, these works limit their training and evaluations to downscaled versions of Phototourism, which makes their results not directly comparable with ours.

**Priors in neural representations.** The integration of pre-trained priors for downstream tasks has emerged as a prominent trend, as they enable the effective incorporation of extrinsic knowledge into diverse applications. For neural representations, priors have been utilized for few-shot scene modeling (Yu et al., 2021; Jain et al., 2021; Deng et al., 2023; Gao et al., 2024) as well as generative tasks (Shue et al., 2023; Poole et al., 2023; Watson et al., 2023; Gao et al., 2024) for object generation and novel view synthesis. In this realm, denoising diffusion probabilistic models (Ho et al., 2020; Rombach et al., 2022) have recently gained particular attention for their application as plug-and-play priors (Graikos et al., 2022). They have been utilized in various domains such as super-resolution (Wang et al., 2023a) and more specifically novel view synthesis. Liu et al. (2023) fine-tune Stable Diffusion (Rombach et al., 2022), a pre-trained latent diffusion model for 2D images, to learn camera controls over a 3D dataset and thus performing NVS by generalizing to other objects. These results hold paramount value, as they highlight the rich 3D prior learned by Stable Diffusion, even though it has only been trained on 2D images. This however comes with geometric inconsistency issues across views, as a pre-trained model alone has no explicit multi-view geometric constraints. In this work, we aim to leverage a pre-trained model to enhance a volumetric scene representation, which inherently adheres to consistency constraints, hence mitigating multi-view consistency issues. The utilization of such priors to enhance in-the-wild scene representation has been an unexplored area of research.

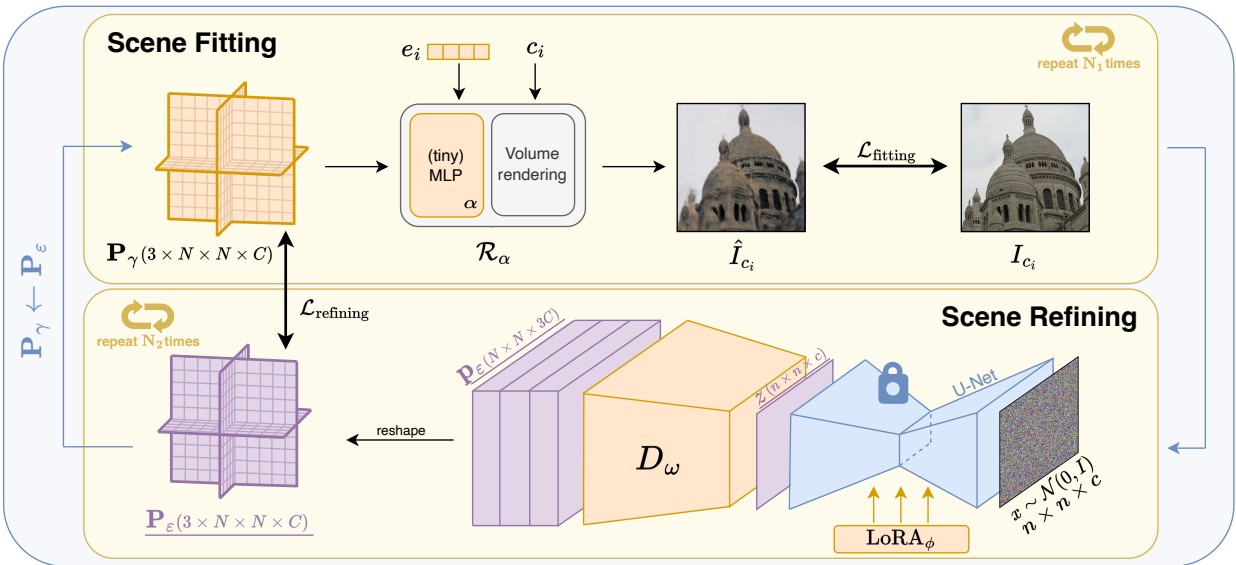

Figure 2: **Scene learning procedure.** The K-Planes $\mathbf{P}_\gamma$, the MLP with trainable parameters $\alpha$, and the appearance embeddings $e_i$ are learned during scene fitting. The LoRA parameters $\phi$ as well as the decoder $D_w$ are learned during scene refining. The pre-trained U-Net is frozen. Assets in violet and underlined are intermediate results. At each iteration, new planes $\mathbf{P}_\varepsilon$ are inferred and assigned to $\mathbf{P}_\gamma$, which are then corrected by scene fitting.

## 3 Method

To guide the optimization of planar scene representations with extrinsic signals, we learn a scene through two alternating stages, as illustrated in Figure 2. *Scene fitting* optimizes our K-Planes representation $\mathbf{P}_\gamma$ to reproduce the images in the training set, as traditionally done in neural rendering techniques. *Scene refining* finetunes a pre-trained network to this K-Planes representation, and then infers a new one $\mathbf{P}_\varepsilon$, which will subsequently be corrected by scene fitting. The main idea behind this is that we use our 3D implicit model $\mathbf{P}_\gamma$ for optimizing the scene to the available information in the training set and adhere to essential geometric constraints, and then project this scene representation on the set of scenes inferable by the pre-trained network, making it closer to natural images. In this section, we detail each stage and elucidate the intuition behind our method.

### 3.1 Scene Fitting

The goal at this stage is to fit a scene, adhering to pre-defined geometric constraints, from posed RGB images. To fit the scene, we adopt the K-Planes representation (Fridovich-Keil et al., 2023). As such, this stage corresponds to optimizing K-Planes to fit a scene, from which we adapt the code.

K-Planes are compact 3D model representations applicable to static scenes, "in-the-wild" scenes (scenes with varying appearances), and dynamic scenes. These models allow for fast training and rendering, while maintaining low-memory usage. K-Planes model a d-dimensional scene with $k = \binom{d}{2}$ planes, which represent the combinations of every pair of dimensions. This structure makes K-Planes compatible with a multitude of neural network architectures, and more particularly image-specialized network architectures. This enables K-Planes inference by minimally tweaking image architectures. For a static 3D scene, $k = 3$ and the planes represent the $xy, xz$, and $yz$ planes. These planes, each of size $N \times N \times C$, encapsulate features representing the density and view-dependent colors of the scene.

The K-Planes model $\mathbf{P}_\gamma$ is originally randomly initialized. The first goal of scene fitting is then to correct this random initialization to fit the training set. Note that the first iteration of scene fitting is especially

particular, since it is starting with a randomly initialized scene, as opposed to a *proposed* scene, as we describe in Section 3.2.

To render the 3D scene from K-Planes, as done by Mildenhall et al. (2020) and Fridovich-Keil et al. (2023), we cast rays from the desired camera position through the coordinate space of the scene, on which we sample 3D points. We decode the corresponding RGB color for each 3D point $\mathbf{q} = (i, j, k)$ by normalizing it to $[0, N)$ and projecting it onto the $k = 3$ planes, denoted as $\mathbf{P}_\gamma^{(xy)}, \mathbf{P}_\gamma^{(xz)}, \mathbf{P}_\gamma^{(yz)}$:

$$f^{(h)}(\mathbf{q}) = \psi(\mathbf{P}_\gamma^{(h)}, \pi^{(h)}(\mathbf{q})) \ , \tag{1}$$

where $h \in \mathbf{H} = \{xy, xz, yz\}$, $\pi^{(h)}(\mathbf{q})$ projects $\mathbf{q}$ onto $\mathbf{P}_\gamma^{(h)}$, and $\psi$ denotes bilinear interpolation on a regular 2D grid.

These features are then aggregated using the Hadamard product to produce a single feature vector of size $M$:

$$f(\mathbf{q}) = \prod_{h \in \mathbf{H}} f^{(h)}(\mathbf{q}) \ . \tag{2}$$

To decode these features, we adopt the hybrid formulation of K-Planes (Fridovich-Keil et al., 2023). Two small Multi-Layer Perceptrons (MLPs), $g_\sigma$ and $g_{RGB}$, map the aggregated features as follows:

$$\sigma(\mathbf{q}), \hat{f}(\mathbf{q}) = g_\sigma(f(\mathbf{q})) \ ,$$
$$c(\mathbf{q}, \mathbf{d}) = g_{\text{RGB}}(\hat{f}(\mathbf{q}), \gamma(\mathbf{d})) \ , \tag{3}$$

where $\gamma(p) = (\sin(2^0 \pi p), \cos(2^0 \pi p), \ldots, \sin(2^{L-1}\pi p), \cos(2^{L-1}\pi p))$ is the positional embedding of $p$. $g_\sigma$ maps the K-Planes features into density $\sigma$ and additional features $\hat{f}$. Subsequently, $g_{\text{RGB}}$ maps $\hat{f}$ and the positionally-encoded view directions $\gamma(\mathbf{d})$ into view-dependent RGB colors. This enforces densities to be independent of view directions.

These decoded RGB colors are then used to render the final image thanks to ray marching and integrals from classical volume rendering (Kajiya & Herzen, 1984), that are practically estimated using quadrature:

$$\hat{C}(\mathbf{r}) = \sum_{i=1}^{N} T_i(1 - \exp(-\sigma_i \delta_i))c_i \ ,$$
$$T_i = \exp\left(-\sum_{j=1}^{i-1} \sigma_j \delta_j\right) \ , \tag{4}$$

where $\hat{C}(\mathbf{r})$ is the expected color, $T_i$ is the accumulated transmittance along the ray, and $\delta_i = t_{i+1} - t_i$ is the distance between adjacent samples.

## 3.2 Scene Refining

This section presents the core of our method, which consists of proposing better optimization initializations for the scene fitting stage. Given a fitted scene representation $\mathbf{P}_\gamma$, this stage consists of learning this fitted implicit representation and proposing a new *refined* representation $\mathbf{P}_\varepsilon$. Formally, this stage consists of projecting our K-Planes $\mathbf{P}_\gamma$ on the set $\mathbb{Q}$ of K-Planes inferable by a low-rank fine-tuning of the pre-trained model:

$$\mathbf{P}_\varepsilon = \underset{P \in \mathbb{Q}}{\arg\min} \ \|\mathbf{P}_\gamma - P\|_2^2 \ . \tag{5}$$

As K-Planes feature channels show similar structure to images (Appendix B), this projection pushes the K-Planes to be even more similar in structure to real images, more particularly to orthogonal projections of the scene on the planes. Figures 5 to 7 illustrate a comparison between feature planes at the end of our optimization and those of standard K-Planes. RefinedFields leads to feature planes exhibiting sharper details, which ultimately lead to more refined details in scene renderings, as proven by our experiments (Figure 3 and Table 3).

---

**Algorithm 1** Alternating training algorithm.

---

1: **Input:** $N_{\text{epochs}}$, $N_1$, $N_2$, $N$, $C$, $n$, $c$, $\mathcal{I} = \{I_{c_i}, c_i\}$, $\mathcal{R}_\alpha$, $\mathbf{D}_w$, $\mathbf{SD}_\phi$, optimizer
2: $x \leftarrow$ standard-gaussian$(n, n, c)$
3: $\mathbf{P}_\gamma \leftarrow$ standard-gaussian$(N, N, 3C)$
4: **for** $N_{\text{epochs}}$ steps **do**
5:     *// scene fitting*
6:     **for** $N_1$ steps **do**
7:         $\gamma, \alpha \leftarrow$ optimizer.step$(\mathcal{L}_{\text{fitting}}(\mathbf{P}_\gamma, \mathcal{I}))$
8:     **end for**
9:     *// scene refining*
10:     **for** $N_2$ steps **do**
11:         $\mathbf{P}_\varepsilon \leftarrow \mathbf{D}_w(\mathbf{SD}_\phi(x))$
12:         $\omega, \phi \leftarrow$ optimizer.step$(\mathcal{L}_{\text{refining}}(\mathbf{P}_\varepsilon, \mathbf{P}_\gamma))$
13:     **end for**
14:     $\mathbf{P}_\gamma \leftarrow \mathbf{P}_\varepsilon$
15: **end for**

---

To provide scene refining with a rich prior, we employ a large-scale pre-trained latent diffusion model, as these networks exhibit great performances as priors for downstream tasks, and share similar properties to our planar representation, both in terms of shape and distribution (Appendix B). More particularly, we adopt Stable Diffusion (Rombach et al., 2022, SD) for its proven performances for downstream 3D (Liu et al., 2023) and 2D (Wang et al., 2023a) tasks. Thus, we integrate the U-Net $\mathbf{SD}_\phi$ and the decoder $\mathbf{D}_\omega$ into our pipeline, and treat the K-Planes as $3C$-channel $N \times N$ images. We also replace the last layer of the decoder $\mathbf{D}_\omega$ with a randomly initialized convolutional layer (with no bias), to take into account the shape of the K-Planes. We then fine-tune the pre-trained model using the fitted K-Planes $\mathbf{P}_\gamma$, and infer refined K-Planes $\mathbf{P}_\varepsilon = \mathbf{D}_w(\mathbf{SD}_\phi(x))$ where $x \sim \mathcal{N}(0, I)$ is sampled once at the beginning of the training. Note that this is different from the multi-step generation process of diffusion model inference, as we only apply the inference at the last time-step of our diffusion model. This is key as our goal here is not to learn distributions over scenes and sample them for generation, but to adapt the pre-trained network and leverage the information already learned within its weights to infer K-Planes closest to representing the scene at hand.

To achieve the fine-tuning of our pre-trained network, a significant challenge presents itself: due to the sheer size of Stable Diffusion, it would be too costly to fine-tune all of its trainable parameters. Moreover, as we only want to modulate priors embedded into the pre-trained network, we look for an alternative to doing full fine-tuning. To circumvent these constraints, we adopt Low-Rank Adaptation (Hu et al., 2022, LoRA), a simple yet effective parameter-efficient fine-tuning method that has proven great transfer capabilities across modalities and tasks (Fan et al., 2023; Lee et al., 2023; Zeng & Lee, 2023). LoRA's relatively minimal design works directly over weight tensors, which means that it can be seamlessly applied to most model architectures. Furthermore, LoRA does not add any additional cost at inference, thanks to its structural re-parameterization design. To achieve this, Hu et al. (2022) inject trainable low-rank decomposition matrices into each layer of a frozen pre-trained model. Let $\mathbf{W}_0$, $\mathbf{b}_0$ be the frozen pre-trained weights and biases, and $x$ be the input. Fine-tuning a frozen linear layer $f(x) = \mathbf{W}_0 x + \mathbf{b}_0$ comes down to learning the low-rank decomposition weights $\Delta\mathbf{W} = \mathbf{BA}$:

$$f(x) = (\mathbf{W}_0 + \Delta\mathbf{W})x + \mathbf{b}_0 \tag{6}$$

where $\mathbf{W}_0, \Delta\mathbf{W} \in \mathbb{R}^{d \times k}$; $\mathbf{B} \in \mathbb{R}^{d \times r}$; $\mathbf{A} \in \mathbb{R}^{r \times k}$; and the rank $r \ll \min(d, k)$.

Thus, to implement scene refining, we fine-tune the LoRA parameters $\phi$ modulating the pre-trained U-Net, as well as the decoder's parameters $\omega$, on the fitted scene $\mathbf{P}_\gamma$. Subsequently, we query the U-Net with Gaussian noise $x$, decode its intermediary output latent $z$ with $\mathbf{D}_\omega$, and infer $\mathbf{p}_\varepsilon$ that is reshaped into a new *refined* scene $\mathbf{P}_\varepsilon$. Finally, $\mathbf{P}_\varepsilon$ is proposed to *scene fitting* as an improved initialization to be optimized. For an in-depth inspection of the feature planes, we refer the reader to Appendix B.

### 3.3 Training

We define an alternating training procedure rotating between scene fitting and scene refining, as described above, and as illustrated in Figure 2.

For *scene fitting*, we train the K-Planes model as proposed by Fridovich-Keil et al. (2023). We use spatial total variation regularization to encourage smooth gradients. This is applied over all the spatial dimensions of each plane in the representation:

$$\mathcal{L}_{\text{TV}}(\mathbf{P}) = \frac{1}{|C|N^2} \sum_{c,i,j} (\|\mathbf{P}_c^{i,j} - \mathbf{P}_c^{i-1,j}\|_2^2 + \|\mathbf{P}_c^{i,j} - \mathbf{P}_c^{i,j-1}\|_2^2) \ . \tag{7}$$

For scenes with varying lighting conditions (e.g. *in-the-wild* scenes as in the Phototourism dataset (Jin et al., 2020)), an $M$-dimensional appearance vector $e_i$ is additionally optimized for each image. This vector is then passed as input to the MLP color decoder $g_{RGB}$ at the rendering step $\mathcal{R}_\alpha$. Hence, the training objective for scene fitting is written as:

$$\min_{\alpha,\gamma} \mathcal{L}_{\text{fitting}} \triangleq \|\mathcal{R}_\alpha(\mathbf{P}_\gamma, \mathbf{C}) - I_{\mathbf{C}}\|_2^2 + \lambda_{\text{TV}} \mathcal{L}_{\text{TV}}(\mathbf{P}_\gamma) \ , \tag{8}$$

where $\mathcal{R}_\alpha$ represents the K-Planes rendering procedure (i.e. ray marching, feature decoding via a small MLP with trainable parameters $\alpha$, and volume rendering), $\mathbf{P}_\gamma$ are the K-Planes with trainable parameters $\gamma$ and $I_{\mathbf{C}}$ is a ground truth RGB image with camera position $\mathbf{C}$.

As for the *scene refining* phase, we optimize the decoder parameters $w$ as well as the LoRA parameters $\phi$, modulating the frozen U-Net weights, on the fitted scene $\mathbf{P}_\gamma$. Thus, the fine-tuning objective for *scene refining* is written as:

$$\min_{w,\phi} \mathcal{L}_{\text{refining}} \triangleq \|\mathbf{D}_w(\mathbf{SD}_\phi(x)) - \mathbf{P}_\gamma\|_2^2 \ , \tag{9}$$

where $x \sim \mathcal{N}(0, I)$ is fixed during scene refining, $\mathbf{SD}_\phi$ is the frozen Stable Diffusion model modulated by LoRA with trainable parameters $\phi$, and $\mathbf{D}_w$ is the latent K-Planes decoder. After this optimization, $\mathbf{P}_\gamma$ is reassigned as $\mathbf{D}_w(\mathbf{SD}_\phi(x))$ and passed to scene fitting. Note that, thanks to the alternating nature of our training and the absence of bias in the decoder's convolutional layers, this optimization does not overfit the model to produce exactly $\mathbf{P}_\gamma$, which is key as it would lead to resuming scene fitting from exactly the same point.

At the end of the alternating training procedure, we save the refined and corrected representation $\mathbf{P}_\gamma$ for rendering and testing. We refer the reader to Algorithm 1 for an overview of our training procedure.

## 4 Experiments

We start by assessing RefinedFields via an experiment on a case study. We then evaluate RefinedFields on synthetic scenes (Mildenhall et al., 2020) and real-world Phototourism (Jin et al., 2020) scenes, where we showcase the improvements our method exhibits relative to our K-Planes base representation. Quantitative results can be found in Tables 2 and 3, where we report for each experiment the Peak Signal-to-Noise Ratio (PSNR) for pixel-level similarity, the Structural Similarity Index Measure (SSIM) for structural-level similarity, and the Learned Perceptual Image Patch Similarity (Zhang et al., 2018, LPIPS) for perceptual similarity. RefinedFields demonstrates an improved performance compared to K-Planes on the task of novel view synthesis. For a further look, experimental details including hyperparameters and more dataset details can be found in Appendix A. Additional qualitative results on synthetic and real-world scenes are available in Appendix C.

### 4.1 Datasets

We evaluate our method similarly to prior work (Mildenhall et al., 2020; Martin-Brualla et al., 2021; Fridovich-Keil et al., 2023) in novel view synthesis, by adopting the *Real Synthetic 360°* dataset (Mildenhall et al., 2020) for synthetic scenes and the same three scenes of cultural monuments from the Phototourism

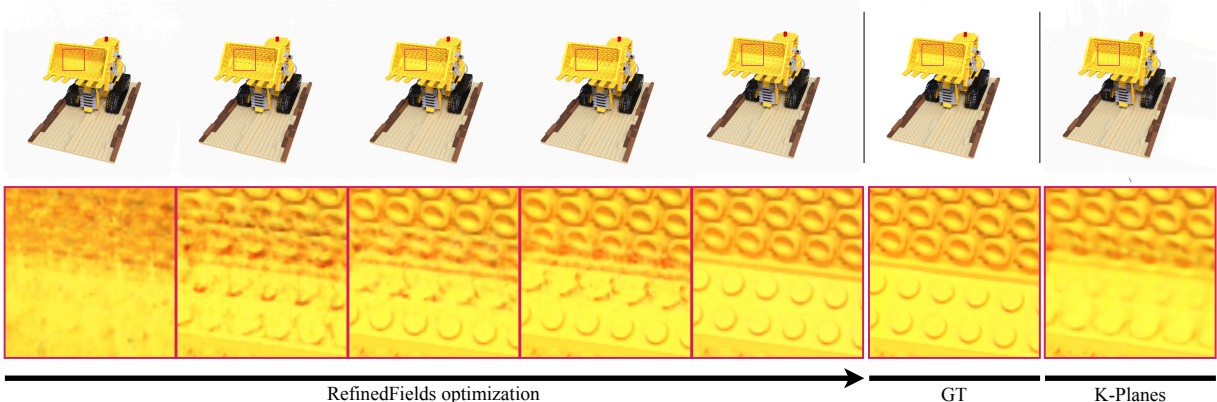

RefinedFields optimization          GT          K-Planes

Figure 3: **Case study.** Qualitative results on the Lego scene from the NeRF synthetic dataset (Mildenhall et al., 2020) showcasing the optimization progression on RefinedFields, and a comparison with the ground truth and K-Planes. The training set is constrained to 50% of its initial size for both RefinedFields and K-Planes. RefinedFields refines the K-Planes representation enabling the proper reconstruction of details in the scene. At the end of optimization, the Mean Squared Error (MSE) for RefinedFields is $3.46 \times 10^{-4}$, while the one for K-Planes is $4.36 \times 10^{-4}$.

Table 2: **Quantitative results.** Results on static synthetic scenes (Mildenhall et al., 2020). The **bold** and underlined entries respectively indicate the best and second-best results. Dashes denote values that were not reported in prior work. Our method outperforms K-Planes, our main baseline, on the task of novel view synthesis for synthetic scenes.

| | PSNR (↑) | | | | | | | | |
| --- | --- | --- | --- | --- | --- | --- | --- | --- | --- |
| | Chair | Drums | Ficus | Hotdog | Lego | Materials | Mic | Ship | Mean |
| NeRF (Mildenhall et al., 2020) | 33.00 | 25.01 | 30.13 | 36.18 | 32.54 | 29.62 | 32.91 | 28.65 | 31.00 |
| TensoRF (Chen et al., 2022a) | 35.76 | 26.01 | **33.99** | **37.41** | 36.46 | **30.12** | 34.61 | 30.77 | 33.14 |
| Plenoxels (Fridovich-Keil et al., 2022) | 33.98 | 25.35 | 31.83 | 36.43 | 34.10 | 29.14 | 33.26 | 29.62 | 31.71 |
| INGP (Müller et al., 2022) | 35.00 | **26.02** | 33.51 | 37.40 | 36.39 | 29.78 | **36.22** | 31.10 | **33.18** |
| K-Planes (Fridovich-Keil et al., 2023) | 34.98 | 25.68 | 31.44 | 36.75 | 35.81 | 29.48 | 34.10 | 30.76 | 32.37 |
| K-Planes-SS (Fridovich-Keil et al., 2023) | 33.61 | 25.27 | 30.92 | 35.88 | 35.09 | 28.83 | 33.01 | 30.04 | 31.58 |
| RefinedFields (ours) | **35.77** | 25.94 | 32.45 | 37.08 | **36.47** | 29.39 | 34.77 | **31.41** | 32.91 |
| | SSIM (↑) | | | | | | | | |
| | Chair | Drums | Ficus | Hotdog | Lego | Materials | Mic | Ship | Mean |
| NeRF (Mildenhall et al., 2020) | 0.967 | 0.925 | 0.964 | 0.974 | 0.961 | 0.949 | 0.980 | 0.856 | 0.947 |
| TensoRF (Chen et al., 2022a) | **0.985** | 0.937 | **0.982** | **0.982** | 0.983 | **0.952** | 0.988 | 0.895 | **0.963** |
| Plenoxels (Fridovich-Keil et al., 2022) | 0.977 | 0.933 | 0.976 | 0.980 | 0.975 | 0.949 | 0.985 | 0.890 | 0.958 |
| INGP (Müller et al., 2022) | — | — | — | — | — | — | — | — | — |
| K-Planes (Fridovich-Keil et al., 2023) | 0.983 | **0.938** | 0.975 | **0.982** | 0.982 | 0.950 | 0.988 | 0.897 | 0.962 |
| K-Planes-SS (Fridovich-Keil et al., 2023) | 0.974 | 0.932 | 0.971 | 0.977 | 0.978 | 0.943 | 0.983 | 0.887 | 0.956 |
| RefinedFields (ours) | **0.985** | 0.937 | 0.980 | 0.981 | **0.984** | 0.945 | **0.989** | **0.903** | **0.963** |

dataset (Jin et al., 2020) for real-world scenes: *Brandenburg Gate*, *Sacré Coeur*, and *Trevi Fountain*. Additional dataset details can be found in Appendix A.1.

## 4.2 Implementation Details

For a fair comparison, we take similar experimental settings in scene fitting to Fridovich-Keil et al. (2023). However, due to the nature of our scene refining pipeline, we limit the implementation in our case to a single-scale K-Planes of $512 \times 512$ resolution, in contrast to the multi-scale approach taken by Fridovich-Keil et al.

(2023) where $N \in \{64, 128, 256, 512\}$. The number of channels in each plane remains the same ($C = 32$). Moreover, throughout all the experiments, we consider the hybrid implementation of K-Planes, where plane features are decoded into colors and densities by a small MLP. As for the scene refining pipeline, we apply no modification to the U-Net in Stable Diffusion. Yet, we replace the last layer of the decoder $D_\omega$ with a new convolutional layer (without bias) to account for the shape of the K-Planes. Hence, the dimensions used in scene refining (Figure 2) are: $N = 512$, $C = 32$, $n = 64$, and $c = 4$. For an in-depth look at our frameworks and hyperparameter settings, we refer the reader to Appendices A.2 and A.3.

### 4.3 Evaluations

**Baselines.** For both synthetic and in-the-wild scenes, we primarily compare our method to our K-Planes baseline, highlighting the improvements it brings to planar scene representations. Note that, for a fair comparison, and to assess the added value of our refining pipeline with respect to our base representation (Section 4.2), we also include a single-scale ablation of K-Planes with $N = 512$ (dubbed K-Planes-SS). In order to provide a more comprehensive perspective, we also illustrate the NVS performances of other recent works that do not employ planar scene representations for synthetic (Mildenhall et al., 2020; Chen et al., 2022a; Fridovich-Keil et al., 2022; Müller et al., 2022) and in-the-wild (Mildenhall et al., 2020; Martin-Brualla et al., 2021; Xu et al., 2024; Kulhanek et al., 2024) scenes.

**Comparisons.** We start by testing our method against K-Planes on a **case study** consisting of the Lego scene from the NeRF synthetic dataset. Here, we train both methods on half of the training set as to deliberately produce a lower-quality fitted scene. As illustrated in Figure 3 our method refines K-Planes and exhibits better quantitative and qualitative results thanks to our scene refining pipeline.

We then apply our method to learn synthetic and in-the-wild scenes. In this case, RefinedFields improves upon not only K-Planes-SS but also K-Planes on the task of NVS (Tables 2 and 3). As a result, it enhances the performance of planar scene representations and brings them closer to recent works, which highlights the value of scene refining. Particularly, for synthetic scenes, RefinedFields improves upon K-Planes and sometimes even outperforms other state-of-the-art methods. For in-the-wild scenes, RefinedFields also improves K-Planes performances in a notable way. However, recent state-of-the-art methods based on Gaussian Splatting architectures (Kerbl et al., 2023) continue to outperform the refined planar representations in terms of overall quality. Figures 1 and 4 show qualitative comparisons of RefinedFields with K-Planes, showing the visual improvements brought by our refining pipeline, which brings finer details to monuments in the Phototourism scenes. Further qualitative results on synthetic and in-the-wild scenes can be found in Appendix C. We also present an inspection of K-Planes features learned by our method and K-Planes in Appendix B.

It is important to note that we do not compare our NVS metrics with other recent works (Chen et al., 2022b; Yang et al., 2023; Zhang et al., 2025). This is because these methods are trained and evaluated on downscaled versions of the Phototourism dataset, which reduces the prominence of fine details in the ground truth for monument structures. As their NVS metrics are computed relative to these ground truth, their results are not comparable to ours. Although other works do compare NVS metrics of various methods across different resolutions, we refrain from this practice as it is not an accurate comparison.

As presented, RefinedFields utilizes an alternating training procedure and a pre-trained prior to refine scene representations, leading to richer details in rendered images. While our method demonstrates promising results, this however comes with a training time increase as compared to our base representation, as our K-Planes feature projection via alternating training leads to the repeated fine-tuning of both scene fitting and scene refining pipelines, which takes overall around 80 hours on a single NVIDIA A100 GPU. We leave the optimization of training time for future work.

### 4.4 Ablations

To justify our choices and explore further, we compare our in-the-wild results (Table 3) to results from two main ablations of our method. **RefinedFields-noFinetuning** is a variation of our method without LoRA fine-tuning. Here, we consider the same exact pipeline (frozen U-Net, same decoder configuration), except that we don't modulate the weights of the frozen U-Net with LoRA. This means that the prior is kept intact

Table 3: **Quantitative results.** Results on three real-world datasets from Phototourism (Jin et al., 2020). Our method shows notable improvements compared to K-Planes on the task of NVS-W. [†]Results from public implementation (Aoi, 2022) reproduced by Fridovich-Keil et al. (2023).

| | Brandenburg Gate | | | Sacré Coeur | | | Trevi Fountain | | |
|---|---|---|---|---|---|---|---|---|---|
| | PSNR (↑) | SSIM (↑) | LPIPS (↓) | PSNR (↑) | SSIM (↑) | LPIPS (↓) | PSNR (↑) | SSIM (↑) | LPIPS (↓) |
| NeRF | 18.90 | 0.8159 | 0.231 | 15.60 | 0.7155 | 0.291 | 16.14 | 0.6007 | 0.366 |
| NeRF-W[†] | 21.32 | — | — | 19.17 | — | — | 18.61 | — | — |
| Splatfacto-W | 26.87 | **0.9320** | **0.124** | 22.53 | **0.8760** | **0.158** | 22.66 | **0.7690** | **0.224** |
| WildGaussians | **27.77** | 0.9270 | 0.133 | **22.56** | 0.8590 | 0.177 | **23.63** | 0.7660 | 0.228 |
| K-Planes | 25.49 | 0.8785 | 0.224 | 20.61 | 0.7735 | 0.265 | 22.67 | 0.7139 | 0.317 |
| K-Planes-SS | 24.48 | 0.8629 | 0.242 | 19.86 | 0.7419 | 0.312 | 21.30 | 0.6627 | 0.355 |
| RefinedFields-noFinetuning | 25.39 | 0.8834 | 0.206 | 21.41 | 0.8059 | 0.239 | 22.54 | 0.7324 | 0.291 |
| RefinedFields-noPrior | 25.42 | 0.8822 | 0.214 | 21.17 | 0.7978 | 0.248 | 22.16 | 0.7251 | 0.291 |
| **RefinedFields (ours)** | 26.64 | 0.8869 | 0.206 | 22.26 | 0.8176 | 0.228 | 23.42 | 0.7379 | 0.284 |

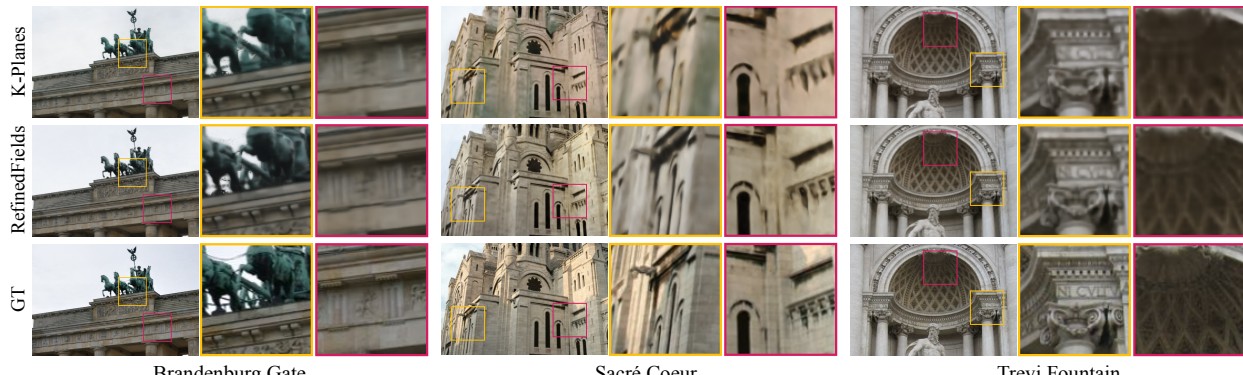

Figure 4: **Qualitative results.** Results on three scenes from Phototourism (Jin et al., 2020). Our method refines K-Planes and leads to richer and finer details in scene renderings.

and no fine-tuning is done. This is to assess the role that LoRA finetuning of the pre-trained prior plays in our pipeline. Note that the decoder is still fine-tuned in this ablation, as additionally freezing it would lead to a random constant optimization initialization $\mathbf{P}_\varepsilon$, which would defeat the purpose of the refining stage. **RefinedFields-noPrior** ablates the prior of Stable Diffusion by randomly re-initializing all U-Net weights while leaving all other elements of the scene-refining pipeline intact. This ablation is done to evaluate the importance of the prior, and to verify that the observed refinements are not entirely resulting from alternate training. Note that ablating the entire scene refining pipeline leads back to the **K-Planes-SS** setting. As illustrated in Table 3, we consistently obtain worse results during the ablation study as compared to our full model, thus demonstrating the value of the pre-trained prior and of LoRA finetuning. This proves the importance of extending scene learning beyond closed-world settings, as our K-Planes projection is done via an optimization conditioning coming from a large image prior that a reasonably sized training set cannot fully capture, especially in-the-wild.

## 5 Conclusion

In this paper, we introduce RefinedFields, a method that refines K-Planes representations by using a pre-trained prior and an alternate training procedure. Extensive experiments show that RefinedFields exhibits notable improvements on the task of novel view synthesis compared to its K-Planes baseline. In concluding this study, several avenues of future work emerge as we consider this work to be a first stepping-stone in improving planar scene representations via conditioning with extrinsic signals. This includes the exploration of approaches to achieve this conditioning other than optimization guidance, and the application of scene refining on other representations.

## Impact Statement

This paper presents work that enhances the construction of high-quality neural representations. As such, the risks associated with our work parallel those of other neural rendering papers. This includes but is not limited to privacy and security concerns, as our method is trained on a dataset of publicly captured images, where privacy-sensitive information (e.g. human faces, license plate numbers) could be present. Hence, similarly to other neural rendering approaches, there is a risk that such data could end up in the trained model if the employed datasets are not properly filtered before use. Furthermore, as our work utilizes Stable Diffusion as prior, it inherits any problematic biases and limitations this model may have.

## Acknowledgments

This work was granted access to the HPC resources of IDRIS under the allocation 2023-AD011014261 made by GENCI. We thank Loic Landrieu, Vicky Kalogeiton and Thibaut Issenhuth for inspiring discussions and valuable feedback.

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

# A    Experimental Details

## A.1    Datasets

**Synthetic dataset.**   For synthetic renderings, we adopt the *Real Synthetic 360°* dataset from NeRF (Mildenhall et al., 2020). This dataset consists of eight path-traced scenes containing objects exhibiting complicated geometry and realistic non-Lambertian materials. Each image is coupled with its corresponding camera parameters. Consistently with prior work, 100 images are used for training each scene and 200 images are used for testing. All images are at $800 \times 800$ pixels.

**In-the-wild dataset.**   For in-the-wild renderings, we adopt the Phototourism dataset (Jin et al., 2020) which is commonly used for in-the-wild tasks. This dataset consists of multitudes of images of touristic landmarks gathered from the internet. Thus, these images are naturally plagued by visual discrepancies, notably illumination variation and transient occluders. Camera parameters are estimated using COLMAP (Schönberger & Frahm, 2016). All images are normalized to [0,1]. We adopt three scenes from Phototurism: *Brandenburg Gate* (1363 images), *Sacré Coeur* (1179 images), and *Trevi Fountain* (3191 images). Testing is done on a standard set that is free of transient occluders.

## A.2    Frameworks

We make use of multiple frameworks to implement our method. Our Python source code (tested on version 3.7.16), based on PyTorch (Paszke et al., 2019) (tested on version 1.13.1) and CUDA (tested on version 11.6), is publicly available as open source. We also utilize Diffusers (von Platen et al., 2022) and Stable Diffusion (tested on version 1-5, main revision). K-Planes also adopt the *tinycudann* framework (Müller, 2021). We run all experiments on a single NVIDIA A100 GPU.

## A.3    Hyperparameters

A summary of our hyperparameters for synthetic as well as in-the-wild scenes can be found in Table 4.

# B    Feature Planes Inspection

In this section, we present a visual inspection of K-Planes feature planes within different contexts. Figures 5 to 7 each correspond to one out of the three orthogonal planes. Each element in Figures 5 to 7 presents a single feature plane, picked randomly from the $C$ feature planes. Figures 5a, 6a and 7a represent the state of the planes at an intermediate stage of the RefinedFields optimization process. Figures 5b, 6b and 7b represent the state of the planes at the end of the RefinedFields optimization. Figures 5c, 6c and 7c represent the planes at the end of the K-Planes-SS optimization (no refining is done in this case).

Two noteworthy observations emerge. First, as seen in Figures 5 to 7, K-Planes feature planes are very similar in structure to real images. In fact, these feature planes depict orthogonal projections of the scene onto the planes. These findings are especially compelling, as they justify the appropriate choice of Stable Diffusion as the pre-trained prior for the refining stage, and provide insight onto the quantitative and qualitative results showcased by our method. Second, a comparison between columns [5b, 6b and 7b] and [5c, 6c and 7c] highlights the impact scene refining has on the feature planes themselves, as planes 5b, 6b and 7b exhibit details that are more similar in structure to the scene, and that are sharper than planes 5c, 6c and 7c.

# C    Supplementary Results

We present below additional qualitative results for in-the-wild scenes (Figures 8 to 10) and synthetic scenes (Figures 11 and 12).

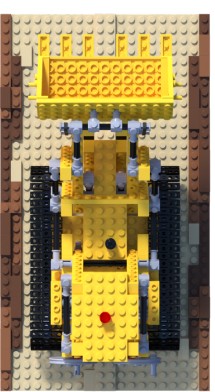

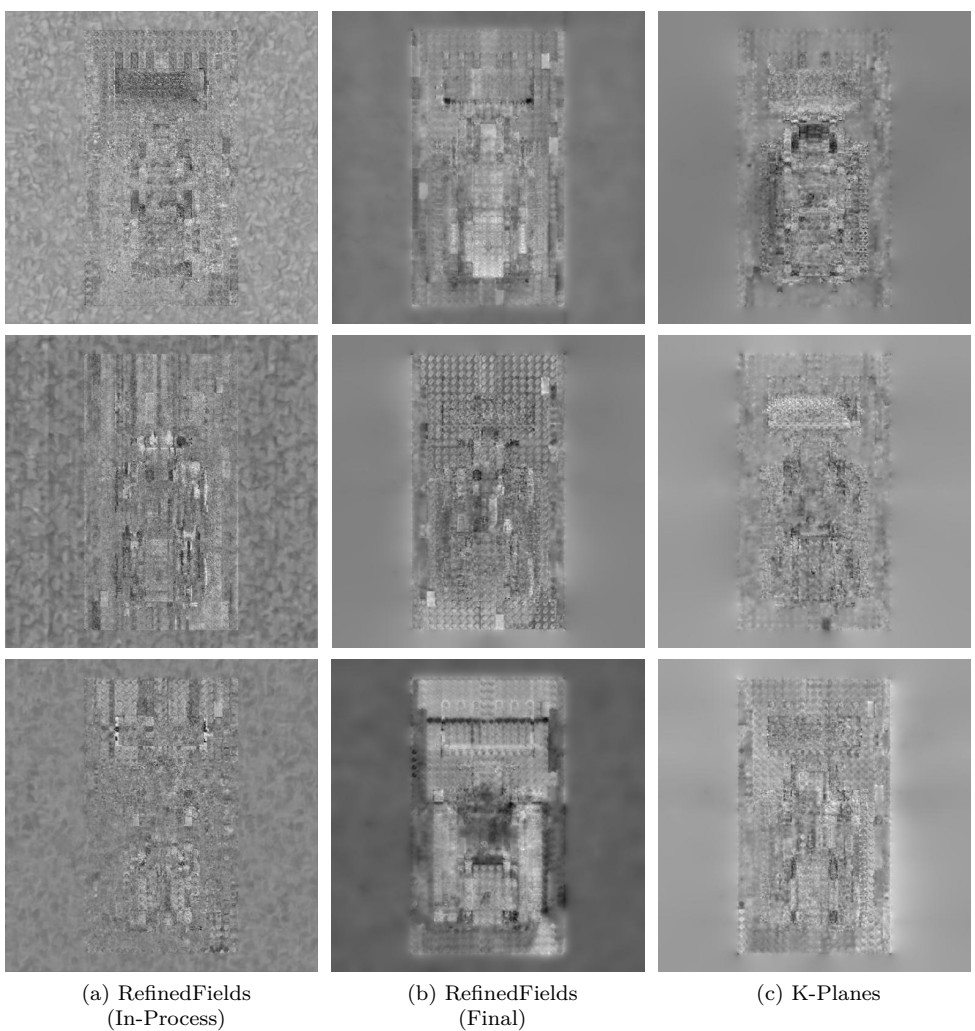

(a) RefinedFields
(In-Process)

(b) RefinedFields
(Final)

(c) K-Planes

Figure 5: **Feature planes inspection.** Visualization of the $(xy)$ K-Planes feature planes during the RefinedFields optimization process (5a), at the end of the RefinedFields optimization (5b), and a comparison with vanilla K-Planes-SS (5c). Feature planes within the $(xy)$ K-Planes are picked randomly.

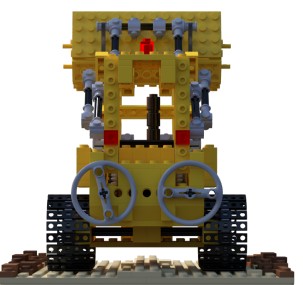

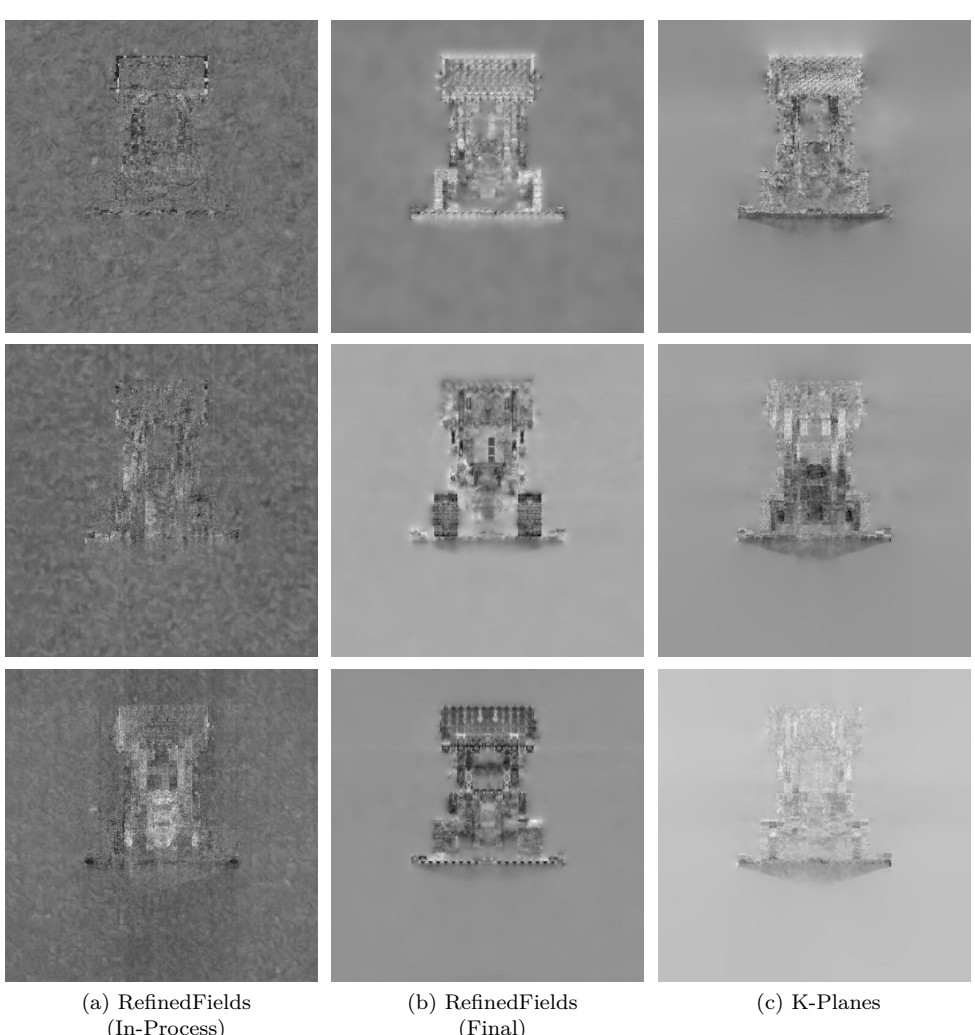

|  (a) RefinedFields  (In-Process) | (b) RefinedFields  (Final) | (c) K-Planes |

Figure 6: **Feature planes inspection.** Visualization of the $(xz)$ K-Planes feature planes during the RefinedFields optimization process (6a), at the end of the RefinedFields optimization (6b), and a comparison with vanilla K-Planes-SS (6c). Feature planes within the $(xz)$ K-Planes are picked randomly.

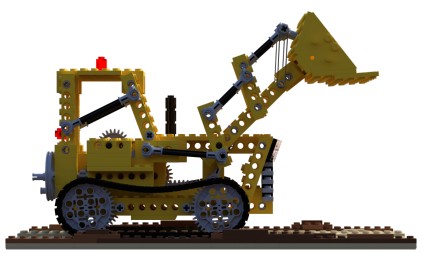

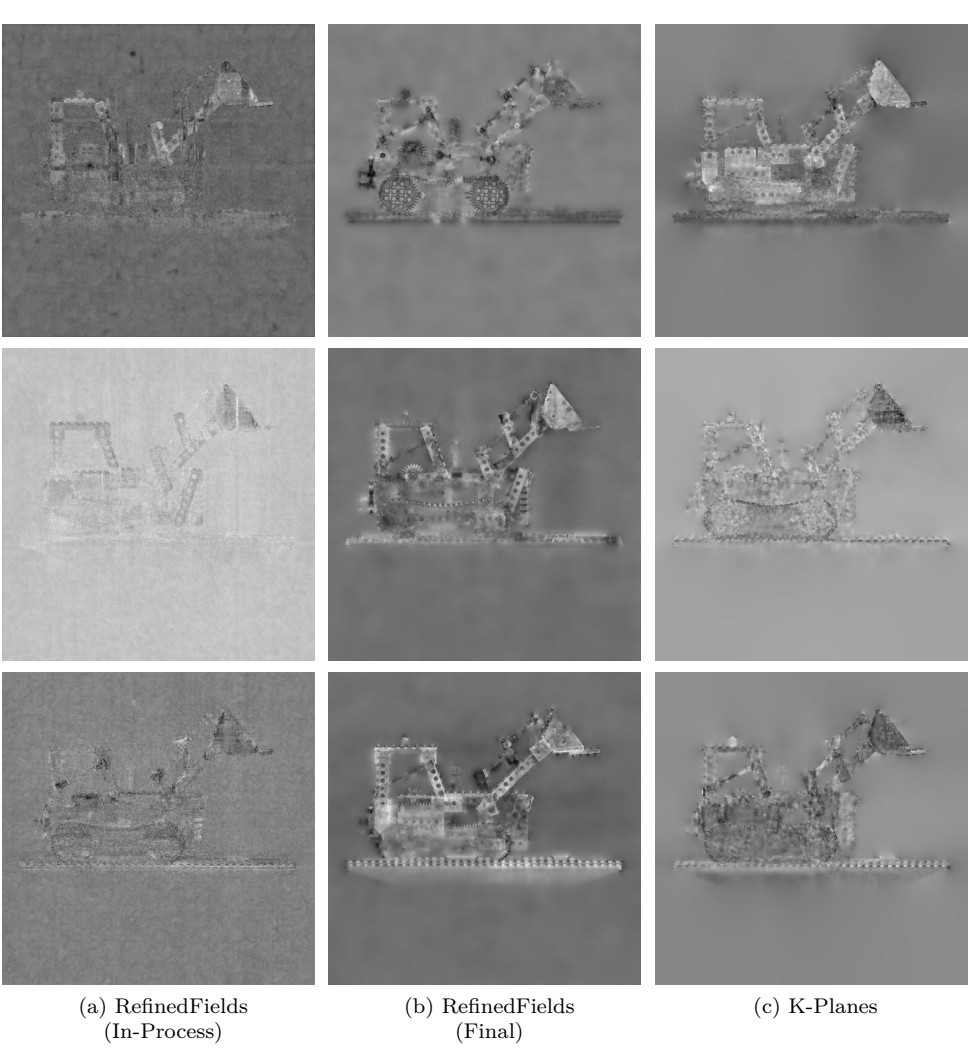

(a) RefinedFields
(In-Process)

(b) RefinedFields
(Final)

(c) K-Planes

Figure 7: **Feature planes inspection.** Visualization of the $(yz)$ K-Planes feature planes during the RefinedFields optimization process (7a), at the end of the RefinedFields optimization (7b), and a comparison with vanilla K-Planes-SS (7c). Feature planes within the $(yz)$ K-Planes are picked randomly.

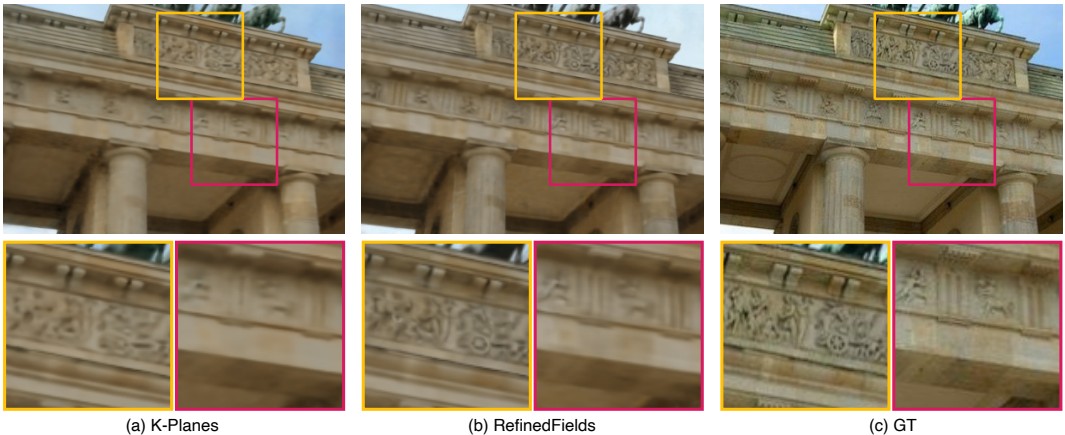

Figure 8: **Qualitative results.** Results on the *Brandenburg Gate* scene from Phototourism (Jin et al., 2020).

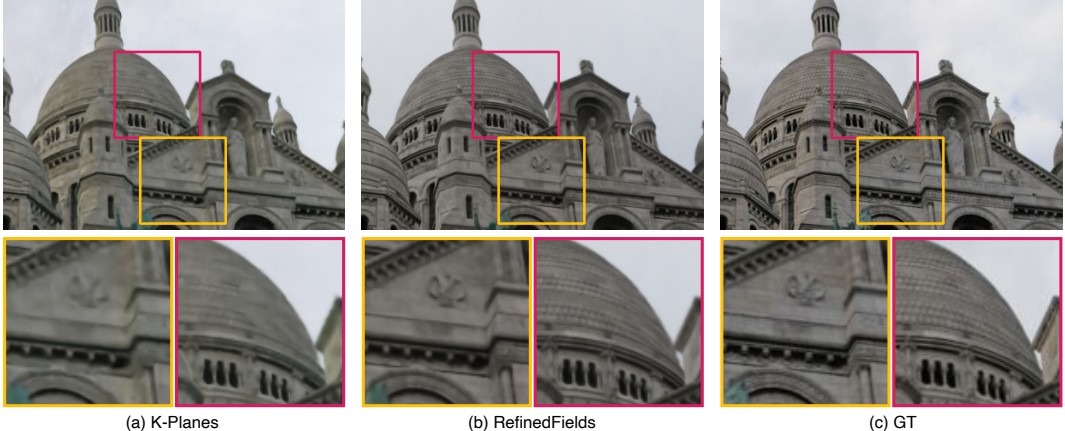

Figure 9: **Qualitative results.** Results on the *Sacré Coeur* scene from Phototourism (Jin et al., 2020).

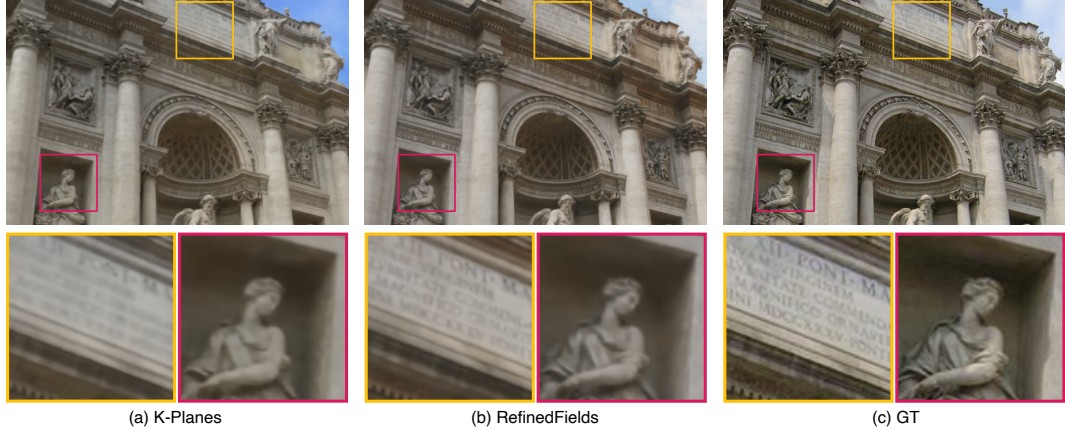

Figure 10: **Qualitative results.** Results on the *Trevi Fountain* scene from Phototourism (Jin et al., 2020).

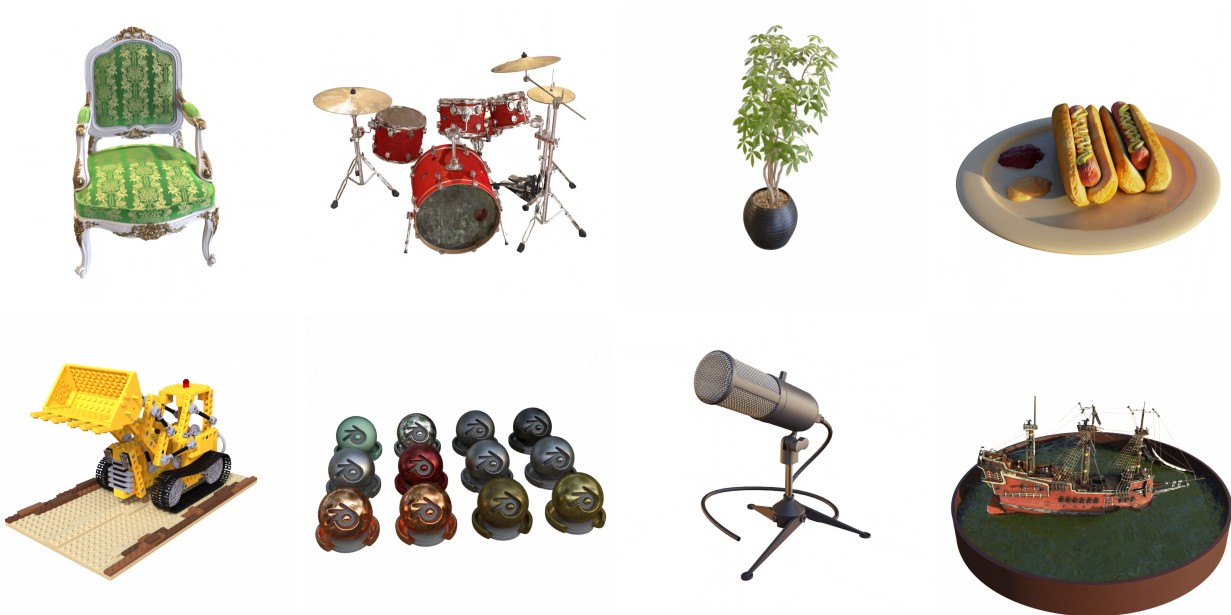

Figure 11: **Qualitative results.** RefinedFields results on the NeRF Synthetic scenes.

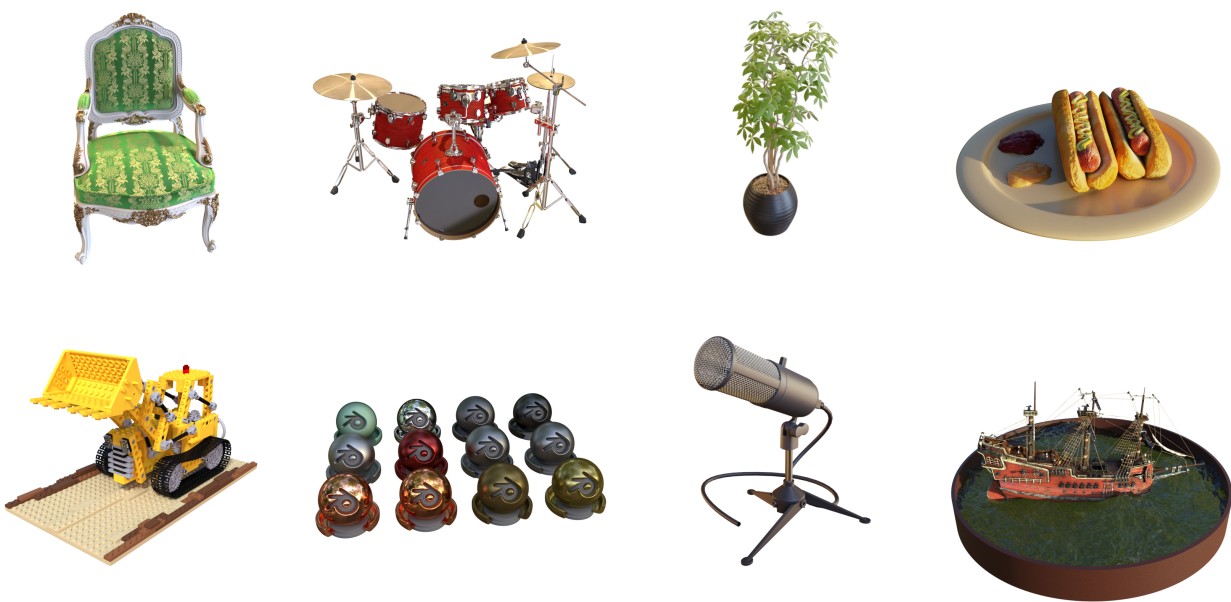

Figure 12: **Ground Truth Renderings.** Ground truth images from the NeRF Synthetic dataset.

Table 4: **Hyperparameters.** A summary of the hyperparameters used to train our model. Appearance optimizations only apply for in-the-wild training.
*Note that this parameter is taken differently from Fridovich-Keil et al. (2023), as we only work with single-scale planes. We consider the highest plane resolution from the multi-scale approach taken by Fridovich-Keil et al. (2023).

| Hyperparameter | Value |
|---|---|
| Epochs ($N_{\text{epochs}}$) | 200 (synthetic) |
| | 20 (*Sacré Coeur*) |
| | 20 (*Brandenburg Gate*) |
| | 10 (*Trevi Fountain*) |
| Fitting iterations ($N_1$) | 30000 |
| Refining iterations ($N_2$) | 3000 |
| Batch size | 4096 |
| Optimizer | Adam |
| Scheduler | Warmup Cosine |
| K-Planes Learning Rate | 0.01 |
| LoRA Learning rate | 0.0001 |
| LoRA rank ($r$) | 4 |
| SD latent resolution | 64 |
| SD channel dimension | 4 |
| SD prompt | " " |
| Number of planes | 3 |
| K-Planes resolution* | 512 |
| K-Planes channel dimension | 32 |
| Epochs Appearance Optimization | 10 |
| Appearance embeddings dimension | 32 |
| Appearance learning rate | 0.1 (*Sacré Coeur*) |
| | 0.1 (*Trevi Fountain*) |
| | 0.001 (*Brandenburg Gate*) |
| Appearance batch size | 512 |

