# OpenReview forum: "RefinedFields: Radiance Fields Refinement for Planar Scene Representations"
_TMLR — Accepted by TMLR_

### Review · Reviewer_ML5J · 2025-03-11

**Summary Of Contributions:**

This paper presents a method to leverage pretrained Stable Diffusion to refine the K-planes representations optimized from a set of input images. Specifically, the authors alternate between training using input images and refining it with pretrained diffusion models to push K-planes features to more closely resemble real-world images. The authors experimented to validate that their methods can improve the vanilla K-Planes models.

**Audience:**

Yes

**Broader Impact Concerns:**

The authors might need to discuss the influence of the proposed method and generative AI on the society.

**Claims And Evidence:**

Yes

**Requested Changes:**

Better clarify the main contributions of the proposed method. Right now it is a simply combination of two stages.

Compare with methods such as CAT3D to demonstrate the effectiveness of the method

Improve quality: the authors might consider applying the same technique to refine more advanced methods, such as WildGaussians, by parametrizing the Gaussians into pixel splat format and compatible with pretrained diffusion models

**Strengths And Weaknesses:**

Strengths:
The idea of refining triplanes with pretrained video models is interesting.
The paper is well-motivated and easy-to-follow.

Weaknesses:
The authors claimed that “This class of methods has not yet been explored for in-the-wild scene modeling, as leveraging priors over representations modeling unconstrained scenes is not evident.” I don’t think this is correct, with the advance of multi-view diffusion models and video generation models, there are a lot of works that use pretrained models to handle novel view synthesis for in-the-wild scenes, such as CAT3D [1]

Another line of work is Dust3R-based methods, which also achieve very high-quality reconstruction results with a feedforward transformer method. These models are trained on several real-world datasets and are assumed to be able to generalize to in-the-wild images.

The results of the proposed model are not good enough: even with refining, the results are still lagging behind existing models such as Splatfacto-W and WildGaussians

Currently, the method is quite simple and involves two quite straightforward stages, I am not quite sure the technical contribution of the methods.

[1] CAT3D: Create Anything in 3D with Multi-View Diffusion Models

---

> ### Author Response · Authors · 2025-03-20
>
> We thank the reviewer for reviewing our paper and for their remarks. We find it important to clarify key concepts that are relevant to the review, in regards to the context of our paper, our claims, and our goals.
>
> ## Referenced related work
>
> The reviewer discusses that some works such as CAT3D [1] and DUSt3R [2] might be related to our method.
> We make the distinction to these works by referencing them in our **paper revision** (introduction (second paragraph), and related work (last paragraph)), and discuss them below.
>
> > [...]  there are a lot of works that use pretrained models to handle novel view synthesis for in-the-wild scenes, such as CAT3D [1].
> >
> > Another line of work is DUSt3R-based methods, which also achieve very high-quality reconstruction results with a feedforward transformer method.
>
> While these methods do tackle scene modeling, they lie in distinctively different lines of research.
>
> - CAT3D models only real scenes (that are captured under some constraints) and **not in-the-wild scenes**, and tackles novel view synthesis in the **few-view setting**, which is a more ill-posed problem, through a **generative approach**. In contrast, and in line with previous in-the-wild methods [3,4], our method focuses on in-the-wild scenes from Phototourism [5], and reconstructs a scene from many captured images. As such, CAT3D is not comparable with our method, as it tackles a different type of scenes in a different problem setting. We discuss the differences between real scenes and in-the-wild scenes, and between reconstructive NVS and generative (few-view) NVS in a separate answer below.
>
> - The referenced DUSt3R [2] approach **does not use implicit scene representations**, and thus lies in a different research direction from our work. This distinction is **explicitly recognized by the DUSt3R authors** themselves, as (i) they do not compare their approach with NeRF-based methods, (ii) they only reference NeRF-based implicit methods in the *Extended Related Work* section to emphasize the differences between their approach and such methods (Appendix C: 3D Reconstruction from Implicit Models):
>
>     > [...] Innovations like Nerf and its followups have pioneered density-based volume rendering to represent scenes as continuous 5D functions. [...] In contrast to the implicit 3D reconstruction, our work focuses on the explicit 3D reconstruction [...]
>
>     In this work, we tackle reconstructing a 3D scene by representing it through an **implicit model** (K-Planes). We compare our work with other approaches also utilizing implicit models.
>
> ## Claim: Priors on representations modeling unconstrained scenes
>
> The reviewer questions the correctness of our claim (in the introduction, end of third paragraph):
> > This class of methods has not yet been explored for in-the-wild scene modeling, as leveraging priors over representations modeling unconstrained scenes is not evident.
>
> by mentioning CAT3D [1] and DUSt3R [2] as related works. As discussed above, **these methods do not lie in the same research direction** and are not related to our work.
>
> Considering the reviewer's remark, **we suggest a refined version of our claim in the paper revision**, which we believe is clearer:
>
> > This class of methods has not yet been explored to enhance implicit models representing in-the-wild scenes, as leveraging priors over such representations is not evident.
>
> We thank the reviewer for bringing this to our attention.

---

> ### Author Response · Authors · 2025-03-20
>
> ## Goal and results: Enhancing the quality of **planar** scene representations
>
> > Improve quality [...]
>
> Regarding the quality of our method, **our results are in-line with our claims**: we bring **notable improvements** to planar scene representations, making their rendering quality **much closer to more advanced state-of-the-art methods**. We recall our goals and results below.
>
> ### Goals
>
> Our main goal is to enhance the quality of current planar scene representations, which have fallen behind state-of-the-art methods leveraging technologies such as gaussian splatting. This is motivated in the introduction and related work of our paper, where we highlight the importance of planar scene representations in the literature, as their planar structure allows for an inter-operability with image-based architectures. **Our claim is that our method enhances K-Planes, and brings their rendering quality closer to current state-of-the-art methods** that are non-planar.
>
> ### Results
>
> As presented in our experiments, **our results confirm our claims**, and bring notable improvements to planar scene representations, outperforming its K-Planes-SS baseline, as well as K-Planes. Particularly, our method brings the rendering quality of our trained planar structures much closer to the current state-of-the-art, which is non-planar (20.61, 22.26, and 22.56 of PSNR for K-Planes, our method, and WildGaussians [7] respectively, on the Sacré Coeur scene).
>
> ## Contribution
>
> > Currently, the method is quite simple and involves two quite straightforward stages, I am not quite sure the technical contribution of the methods.
>
> We recall the contributions of our work below.
> - Our work starts from the observation that planar structures exhibit features similar to real images.
> - Our contribution leverages this insight and proposes a novel algorithm that utilizes pre-trained image priors to enhance the quality of planar scene representations. It consists of proposing a new pipeline, which we call *scene refining*, that results in a better optimization initialization for K-Planes.
> - Additionally, as we clarify in our claim, such an application for pre-trained image priors has not been previously explored for learning in-the-wild scenes using implicit representations.
>
> In our paper revision, we clarify our contribution in the introduction (RefinedFields proposal) and in section 3.2.
>
> While the reviewer describes our approach as “quite simple,” we consider this to be subjective. Moreover, simplicity does not diminish the soundess of a contribution, nor its novelty or impact. Leveraging pre-trained image priors to enhance planar scene representations offers a practical solution, particularly in the context of implicit representations for in-the-wild scenes.
>
> ## Broader Impact Concerns
>
> We added the following impact statement in our paper revision.
> > This paper presents work that enhances the construction of high-quality neural representations. As such, the risks associated with our work parallel those of other neural rendering papers. This includes but is not limited to privacy and security concerns, as our method is trained on a dataset of publicly captured images, where privacy-sensitive information (e.g. human faces, license plate numbers) could be present. Hence, similarly to other neural rendering approaches, there is a risk that such data could end up in the trained model if the employed datasets are not properly filtered before use. Furthermore, as our work utilizes Stable Diffusion as prior, it inherits any problematic biases and limitations this model may have.

---

> ### Author Response · Authors · 2025-03-20
>
> ## In-the-wild vs real scenes.
>
> - **Real scenes** refer to scenes taken under some constraints (e.g. static scene, constant lighting). Many current works tackle the problem of novel-view synthesis on real scenes, including the referenced CAT3D used with the *sparse multi-view to 3D* modality.
> - **In-the-wild scenes** refer to scenes that exhibit no constraints: they include high variation in lighting (e.g. sunny, cloudy, night) and transient occluders (e.g. moving cars, pedestrians, obstructing posters on monuments). Particularly, the Phototourism dataset [5], has become the go-to dataset in recent works tackling in-the-wild scene modeling.
>
> As discussed, our work tackles **in-the-wild scenes**, and we compare it with other in-the-wild methods.
>
>
> ## Scene reconstruction vs (few-view) generation
>
> - **Reconstructive NVS** consists in reconstructing a scene from its captured images. NeRF [6], NeRF-W [3], K-Planes [4], and our method are all scene reconstruction methods.
> - **Generative NVS** generates novel views of a scene (e.g. by utilizing a diffusion model), and uses them to model the scene. This is typically done in few-view settings, where only a few captured images are available (which is the setting of CAT3D [1]).
>
> As discussed, our work tackles **reconstructive NVS** (for in-the-wild scenes). We thus compare it with other works lying in the same context.
>
> ## References
>
> [1] Gao, R., Holynski, A., Henzler, P., Brussee, A., Martin-Brualla, R., Srinivasan, P. P., … Poole, B. (2024). CAT3D: Create Anything in 3D with Multi-View Diffusion Models. Advances in Neural Information Processing Systems.
>
> [2] Wang, S., Leroy, V., Cabon, Y., Chidlovskii, B., & Revaud, J. (2024, June). DUSt3R: Geometric 3D Vision Made Easy. Proceedings of the IEEE/CVF Conference on Computer Vision and Pattern Recognition (CVPR), 20697–20709.
>
> [3] Martin-Brualla, R., Radwan, N., Sajjadi, M. S. M., Barron, J. T., Dosovitskiy, A., & Duckworth, D. (2021, June). NeRF in the Wild: Neural Radiance Fields for Unconstrained Photo Collections. Proceedings of the IEEE/CVF Conference on Computer Vision and Pattern Recognition (CVPR), 7210–7219.
>
> [4] Fridovich-Keil, S., Meanti, G., Warburg, F. R., Recht, B., & Kanazawa, A. (2023, June). K-Planes: Explicit Radiance Fields in Space, Time, and Appearance. Proceedings of the IEEE/CVF Conference on Computer Vision and Pattern Recognition (CVPR), 12479–12488.
>
> [5] Jin, Y., Mishkin, D., Mishchuk, A., Matas, J., Fua, P., Yi, K. M., & Trulls, E. (2020). Image Matching Across Wide Baselines: From Paper to Practice. International Journal of Computer Vision, 129(2), 517–547.
>
> [6] Mildenhall, B., Srinivasan, P. P., Tancik, M., Barron, J. T., Ramamoorthi, R., & Ng, R. (2020). NeRF: Representing Scenes as Neural Radiance Fields for View Synthesis. ECCV.
>
> [7] Kulhanek, J., Peng, S., Kukelova, Z., Pollefeys, M., & Sattler, T. (2024). WildGaussians: 3D Gaussian Splatting In the Wild. The Thirty-Eighth Annual Conference on Neural Information Processing Systems.

---

### Review · Reviewer_etk7 · 2025-03-11

**Summary Of Contributions:**

This paper extends K-Planes leveraging pre-trained generative models as a rich prior to refine planar scenes representations. The training procedure iteratively switches between optimizing a K-Planess representation on images from a particular dataset (scene fitting), and fine-tuning a pre-trained network to output a new conditioning leading to a refined version of this K-Planes representation (scene refining).

**Audience:**

Yes

**Broader Impact Concerns:**

No concerns.

**Claims And Evidence:**

Yes

**Requested Changes:**

Please see the weaknesses.

**Strengths And Weaknesses:**

### Strengths

- The proposed approach guides volumetric scene modeling by leveraging not only the training dataset at hand but also the rich prior lying within the weights of the pre-trained model.
- The paper is well-written and the results are promising, with an evaluation on synthetic and real scenes.

### Weaknesses

- The proposed method is really interesting, however its scope seems limited only to K-Planes. Can RefinedFields be applied also to other models?

- The proposed method increases marginally and partially K-Planes results (Table 2), but what is the computational overhead? How do you choose the number of fitting and refining iterations?

- (Minor) Best results should be highlighed in Table 3 and the arrows in the metrics header are missing.

---

> ### Author Response · Authors · 2025-03-20
>
> We thank the reviewer for reviewing our paper and for their remarks. We address below the questions raised by the reviewer.
>
> > Can RefinedFields be applied also to other models?
>
> RefinedFields could be applied to any planar representation. We utilize K-Planes in our work as their planar structure makes them compatible with image-based methods. This is a key property which made them prominent in the literature.
>
> > The proposed method increases marginally and partially K-Planes results (Table 2), but what is the computational overhead? How do you choose the number of fitting and refining iterations?
>
> We consider our results to be notable, as our method brings the rendering quality of our trained planar structures much closer to the current state-of-the-art (20.61, 22.26, and 22.56 of PSNR for K-Planes, our method, and WildGaussians respectively, on the Sacré Coeur scene).
>
> As discussed in Section 4.3, RefinedFields currently takes around 80 hours on a single NVIDIA A100 GPU. The number of fitting and refining iterations is chosen empirically so that the alternating training algorithm switches phases as soon as it reaches convergence. We present these hyperparameters in Table 4.
>
> > Table 3
>
> We appreciate the reviewer’s remarks regarding Table 3, which we implement in our paper revision.

---

### Review · Reviewer_GN77 · 2025-03-12

**Summary Of Contributions:**

This paper introduces RefinedFields, a novel method that enhances  the quality of the K-Planes representation by integrating pre-trained networks. The key contribution is a training scheme that iterates between scene fitting (optimizing K-Planes with training images) and scene refining (leveraging a pre-trained model to enhance the representation). In this way, the method can leverage the prior information in the pre-trained diffusion model to regularize the K-Planes representation. This approach enables richer details in novel view synthesis (NVS) and improves in-the-wild scene modeling. Extensive experiments demonstrate that RefinedFields significantly outperforms K-Planes.

**Audience:**

Yes

**Claims And Evidence:**

Yes

**Requested Changes:**

Please refer to weaknesses.

**Strengths And Weaknesses:**

**Strengths**:

1.This idea is interesting and presents a novel approach to leveraging prior information for novel view synthesis. It utilizes only existing pre-trained models without requiring additional training data, making it both efficient and scalable for future applications.

2.The paper is well-written, and the experiments effectively demonstrate the method's effectiveness.

**Weaknesses**:

1.The pre-trained diffusion model is not trained in the feature plane domain, and the reviewer is unsure what kind of prior is used in the proposed method. Some toy experiments or additional discussion may be needed to clarify this point.

2.In the ablation studies, "RefinedFields-noFinetuning" refers to the method without LoRA on U-Net. However, it is unclear whether the decoder is still fine-tuned. Clarification on this aspect would be helpful.

---

> ### Author Response · Authors · 2025-03-20
>
> We thank the reviewer for reviewing our paper and for their remarks. We address below the questions raised by the reviewer.
>
> > 1. The pre-trained diffusion model is not trained in the feature plane domain [...]
>
> While the pre-trained diffusion model is trained on images and not feature planes, we show in Appendix B that the contents of feature planes actually resemble projected images of the scene, a key insight on which our method is built. This observation is also noted in [1] (Figure 2). Moreover, we include in Appendix B a comparison between the feature planes trained with RefinedFields and those of K-Planes, highlighting the impact of our method on these planes, which exhibit sharper details. In our paper revision, we further discuss this and reference the appendix in the main part of the paper (page 6).
>
> > 2. In the ablation studies, [...] it is unclear whether the decoder is still fine-tuned [for "RefinedFields-noFinetuning"].
>
> In RefinedFields-NoFineTuning, we omit the LoRA finetuning of the U-Net, but the decoder is still fine-tuned. This is because additionally freezing the decoder would result in the entire scene refining pipeline being frozen, and thus a random and constant optimization initialization $\mathbf{P}_\varepsilon$ outputted from our refining stage throughout the training. This would make refining ineffective, as it is equivalent to re-running a standard K-Planes optimization. We clarify this in the paper revision (Section 4.4), and thank the reviewer for bringing this to our attention.
>
> ## References
>
> [1] Liu, Y.-T., Guo, Y.-C., Luo, G., Sun, H., Yin, W., & Zhang, S.-H. (2024, June). PI3D: Efficient Text-to-3D Generation with Pseudo-Image Diffusion. Proceedings of the IEEE/CVF Conference on Computer Vision and Pattern Recognition (CVPR), 19915–19924.

---

### Author Response · Authors · 2025-03-20
**Global Response and Revision**

We would like to thank the reviewers for their positive and constructive feedback.

We appreciate that they found the paper to be **well-written** (GN77, etk7) and **easy to follow** (ML5J), **interesting** (GN77, etk7, ML5J), and **scalable for future applications** (GN77). We also value that they found our results **promising** (etk7), effectively **demonstrating our method's effectiveness** (GN77). All reviewers found that **our main claims are properly supported**.

We clarified in our **paper revision** some points raised by the reviewers (highlighted in blue).
- In the **introduction**:
    - *[Reviewer ML5J]* We have **clarified our claim** regarding the use of pre-trained models to enhance implicit models representing in-the-wild scenes. We believe this version is more accurate and eliminates all ambiguity.
    - *[Reviewer ML5J]* We have also **clarified the core of our technical contribution** (under RefinedFields proposal).
- In the **method** section:
    - *[Reviewer ML5J]* We added a sentence that recalls the core technical contribution of our method.
    - *[Reviewer GN77]* We further **justified the use of our pre-trained model** for K-Planes by including a more elaborate discussion that references the appendix section.
- In the **experiments** section:
    - *[Reviewer GN77]* We have clarified the experimental settings of one of our ablations.
    - *[Reviewer etk7]* We improved the formatting of Table 3.
- *[Reviewer ML5J]* We added a new section "**Impact Statement**" that discusses the potential broader impact of our work.

We remain open for further discussions with the reviewers.

---

### Decision · Action_Editor_dsgx · 2025-04-22

**Recommendation:** Accept as is

**Comment:**

While the reviewers acknowledged the interest of the core idea behind the work and the clarity of the paper, they expressed some concerns about the technical contributions and its scope, the relationship with some existing methods, and some aspects of the empirical results. The authors' feedback clarified most of these points, leading to two reviewers leaning to accept, but one leaning to reject. The main argument for rejection is the limited contributions of the work. Considering that this argument goes against the TMLR rules for rejection based on novelty, and that the paper is supported by the other two reviewers, this AE recommends acceptance.

**Audience:**

The three reviewers acknowledge that there is a TMLR audience for this work.

**Claims And Evidence:**

The three reviewers acknowledge that the claims and evidence are supported.